

# A comparison of Structure from Motion Photogrammetry and the Traversing Micro Erosion Meter for measuring erosion on rock shore platforms.

Niamh D. Cullen[1], Ankit K. Verma[1] and  Mary C. Bourke[1]

[1]Department of Geography, Trinity College Dublin, The University of Dublin, 2 College Green, Dublin 2, Ireland.

*Correspondence to*: Niamh D. Cullen (cullenni@tcd.ie)

**Abstract**

For decades researchers have used the Micro Erosion Meter and it successor the Traversing Micro Erosion Meter to measure microscale rates of vertical erosion (downwearing) on rock shore platforms. Difficulties with 'upscaling' of microscale field data in order to explain long term platform evolution have led to calls to introduce other methods which allow measurement of platform erosion at different scales. Structure from Motion Photogrammetry is fast emerging as a reliable, cost-effective tool for geomorphic change detection, providing a

valuable means for detecting micro to meso-scale geomorphic change over different terrain types. Here we present the results of an experiment where we test the efficacy of Structure from Motion Photogrammetry for measuring change on rock shore platforms due to different erosion processes (sweeping abrasion, scratching and percussion). Key to this approach is the development of the Coordinate Reference System used to reference and scale the models, and which can be easily deployed in the field. Experiments were carried out on three simulated platform

surfaces with low to high relative rugosity to assess the influence of surface roughness. We find that a Structure from Motion Photogrammetry can be used to reliably detect micro (sub mm) and meso (cm) scale erosion on shore platforms with a low Rugosity Index. As topographic complexity increases, the scale of detection is reduced. We also provide a detailed comparison of the two methods across a range of categories including cost, data collection, analysis and output. We find that Structure from Motion offers several advantages over the Micro

Erosion Meter, most notably the ability to detect and measure erosion of shore platforms at different scales.

**Keywords**

Shore platforms, Structure from Motion Photogrammetry, Traversing/Micro Erosion Metre, erosion.

## 1 Introduction

There are numerous methods employed for measuring natural rates of change on rock surfaces. For decades researchers were restricted to direct measurement of change relative to a datum, however this method has been



largely superseded by techniques which fall into two general categories; contact methods which utilise erosion meters, and non-contact methods such as Terrestrial Laser Scanning (TLS) and Structure from Motion (SfM) Photogrammetry (Moses et al., 2014). On rock shore platforms, the Micro Erosion Meter (MEM) and its successor

the Transverse Micro Erosion Meter (TMEM) are the most frequently applied instruments for quantifying micro-scale erosion. However, SfM Photogrammetry is fast emerging as a valuable tool for detecting and quantifying geomorphic change across a range of scales and environments and represents a potential alternative to the MEM and TMEM for measuring erosion on shore platforms if a suitable level of resolution, accuracy and repeatability can be achieved. There is a large body of literature focussed on each of these methods (e.g. Carrivick et al., 2016;

Gómez-Gutiérrez et al., 2014; Hanna, 1966; Kaiser et al., 2014; Smith et al., 2016; Snavely, 2006; Stephenson and Finlayson, 2009; Stephenson et al., 2010; Stephenson and Kirk, 2001; Trenhaile, 2006; Trudgill, 1975; Trudgill et al., 1981; Westoby et al., 2012). A brief overview of the two methods is given below.

### 1.1 The Micro Erosion Meter and the Traversing Micro Erosion Meter

The MEM was developed and described by Hanna (1966) and High and Hanna (1970) as a tool for measuring relatively slow lowering rates of bedrock surfaces. Since its inception, the MEM and its modified successor, the TMEM (Trudgill et al., 1981) (hereafter T/MEM) have been used by numerous researchers to measure rates of surface lowering on shore platforms of varying lithologies. The spatial and temporal variability of measured erosion rates for shore platforms have allowed a more detailed understanding of processes operating on shore

platform, contributing to the ongoing debate on the origin of shore platforms and the relative contributions of marine, biological and subaerial processes which drive their evolution (See Stephenson and Finlayson, 2009, for a more detailed review of the contribution of the T/MEM to rock coast research). The popularity of the T/MEM stems from the ability to detect sub-mm changes over short timescales (2 years) which, while comparative with the duration of many research projects, is also considered representative of longer-term (decadal) measurements

(Stephenson et al., 2010). Add to this, the often cited low cost of construction and portability of the instrument and its popularity among rock coast researchers is easily understood.

Moses et al. (2014) outlined some limitations associated with the T/MEM. Authors studying erosion on (relatively soft) chalk platforms noted that the probe might cause erosion of the platform surface. This 'probe erosion' was also noted by Spate (1985). However, this does not constitute a problem where erosion rates are rapid (Foote et

al., 2006; Swantesson et al., 2006). In addition, Moses et al., (2014) also noted that where rapid rates of erosion occur, this may result in the loosening or dislodgement of the bolts on which the T/MEM is placed on annual (Ellis, 1986; Andrews, 2000), or decadal timescales (Stephenson and Kirk, 1966). Trenhaile (2003) noted that although the T/MEM records small amounts of platform downwearing, it cannot record wave quarrying of larger blocks or loss of rock fragments due to frost riving.

Our use of the instrument has identified some additional limitations. First, the location of a T/MEM measurement station is limited to surfaces with low topographic complexity. This is an issue for shore platforms with highly variable meso and macro scale roughness and which only broadly conform to the Sunamura's (1992) traditional Type A and Type B classification. Excluding these more complex platform morphologies significantly limits our ability to quantify rates and identify processes and styles of shore platform erosion across the complete spectrum





of platform morphologies. Second, while decades of measuring micro-scale erosion using the T/MEM have provided valuable insights into rates and processes of downwearing on shore platforms, there are difficulties associated with 'up-scaling' these field data to explain meso and macro-scale landform development (Warke and McKinley, 2011). A recent study that reviewed 95 publications on shore platforms highlighted a concentration of research on micro and macro scale studies. (Cullen and Bourke 2018), also noted by Stephenson and Naylor (2011). In comparison, meso-scale processes have received less attention, although research at this scale has increased significantly in the last two decades (Cullen and Bourke, 2018). Indeed, Stephenson et al. (2010) advocated the introduction of new techniques which capture the full range of scales of erosion on shore platforms. SfM Photogrammetry is one such technique that has this potential.

## 1.2 Structure from Motion Photogrammetry

Significant developments in digital photogrammetry techniques over the last decade have revolutionised the collection of 3D topographic data in the geosciences. Traditional photogrammetry requires a knowledge of the 3D location and orientation of the camera and accurate 3D information of control points in the scene of interest. While methods which allow the accurate calibration of non-metric cameras and reliable automation of the photogrammetric process have enhanced the use of photogrammetry in the geosciences (e.g. Carbonneau et al., 2004; Chandler, 1999; Chandler et al., 2002), it still requires expert understanding and practice (Carrivick et al., 2016). In the last decade, there have been significant workflow advancements which have dramatically reduced the expertise required. Structure from Motion (SfM) photogrammetry, uses a standard camera for collecting image data of a three-dimensional (3D) landform.

Multiple overlapping images are taken from different spatial positions and used to reconstruct the 3D geometry of the target. Unlike traditional photogrammetry, the SfM workflow does not require prior knowledge of the 3D location, the camera orientation or 3D information on control points before reconstruction of scene geometry. Rather, Scale Invariant Feature Detection (SIFT) (Lowe, 2004) is used to match points between images, and a least square bundle adjustment algorithm is used to align images and produce a 'sparse' point cloud representing the most prominent features in the images. A further development utilises Multi-View Stereo (SfM-MVS) algorithms (e.g. Furukawa et al., 2010) to intensify the sparse cloud and merge the resulting 3D point cloud into a single dense point-based model. This can then be used to generate a high-resolution ortho-photo, mesh or Digital Elevation Model (DEM). Successive point clouds and DEMs of the same location or feature can be analysed utilising widely available software (e.g. ArcMap, CloudCompare) for geomorphic change detection to quantify erosion and deposition. A large amount of literature has been published on SfM, and the reader is referred to Carrivick et al. (2016); Fonstad et al. (2013); Micheletti et al. (2015a, 2015b); Özyeşil et al. (2017); Smith et al. (2016); Thoeni et al. (2014); Walkden and Hall (2005); Westoby et al. (2012), Verma and Bourke ( for more detailed discussions of SfM techniques and workflows.

The SfM-MVS workflow has been widely applied in the geosciences at varying scales of resolution from small scale (mm - cm's) scale studies of soil erosion to morphodynamic studies of beaches, coastal cliffs and braided rivers (e.g. Balaguer-Puig et al., 2017; Brunier et al., 2016a; Brunier et al., 2016b; Javernick et al., 2014; Kaiser et al., 2014; Lim et al., 2010). SfM-MVS offers several advantages over traditional surveying techniques,





specifically its relatively low cost and portability of required equipment, i.e. a camera, compared to that of TLS.
In addition, the availability of free and relatively low cost commercial software, a semi-automated workflow and
the decreasing cost of high-end desktop computers have resulted in the increasing application of this method in
geomorphological research.

It is worth noting that the accuracy and resolution of SfM-MVS derived DEMs relies heavily on the quality of the
images used and the accuracy of the coordinate reference system. For work on shore platforms, the accuracy of
the DEM is limited by the accuracy of the Ground Control Points (GCPs) used. These are often determined using
a Differential GPS (dGPS) or total stations which have reported accuracies of centimetres and millimetres
respectively. However, a number of rock breakdown processes, such as granular disintegration (Viles, 2001) and
features, such as weathering pits (Bourke et al., 2007; Thornbush, 2012; Viles, 2001) occur at cm to sub-mm scale.

Our work has three foci: First, to test the SfM-MVS for measuring micro-scale erosion on shore platforms. Second
to determine the potential of SfM-MVS for meso-scale geomorphic change detection . Third, to provide a robust
assessment and comparison of the two methods (T/MEM and SfM-MVS) for measuring erosion on shore
platforms. Key to our approach is to adapt the local coordinate reference system (CRS), and SfM-MVS workflow
developed by Verma and Bourke ( Their system was developed to generate sub-mm scale DEMs of rock surfaces
(<10 m$^2$) in difficult to access terrains (e.g., cliffs and steep-sided impact crater walls). Their method can produce
high resolution (sub-mm) DEMs with sub-mm accuracy. We advance this work through the design and
manufacture of a field-hardy CRS which can be quickly deployed, repeatedly at the same site. Our approach will
enable the application of SfM-MVS for geomorphic change detection on shore platforms at both the micro and
meso scale.

In this paper we present the results of a series of experiments on simulated platform surfaces using our newly
developed CRS.

## 2 Methods

### 2.1 A manufactured Coordinate Reference System for SfM-MVS

We have adapted the local coordinate reference system of (Verma and Bourke)which utilises a precisely measured
equilateral triangle with a coded marker (downloaded from Agisoft Photoscan) attached at each vertex (Figure 1a
and b). The x, y and z coordinates of each coded marker are calculated using trigonometry and serve as the GCPs
for generating the DEMs in the SfM-MVS workflow. When used for a small surface area (≤ 6.76 m$^2$), this method
has been proved to produce high resolution (0.5 mm per pixel) DEMs with sub-mm accuracy (Verma and Bourke).


We mounted the coded markers onto a specifically designed stainless-steel platform (Fig. a and b). The platform
consists of a 15 cm equilateral triangle with three square steel plates (4 cm x 4 cm x 0.5 cm) and a specially





machined leg. Each plate is engineered so that the centre of a plate is fixed precisely (± 0.01mm) on one vertex of the triangular base. The centre of each plate is also permanently marked during manufacture so that the coded markers can be accurately placed. The base of the leg is machined to fit a stainless-steel square head bolt to a depth of 1.5 cm and is fixed at the centre of gravity on the underside of the triangular base plate.

In the field, the square headed bolt is fixed to the platform by drilling a hole and fixing the bolt with marine grade epoxy resin, making sure the bolt head is level. This is similar to the approach used to install T/MEM stations. When mounted onto the bolt, this design secures the base plate with the coordinate system in place with a high degree of relocation precision. This permits repeated measurements and the georeferencing of DEMs for high resolution change detection of field sites.

## 2.2 The experiments

The experiments were designed to capture different scales of erosion fromgranular scale abrasion of the platform surface to the removal of rock fragments. The accuracy of the SfM-MVS generated DEMs used to calculate DEMs of Difference (DoDs) for geomorphic change detection were assessed by means of horizontal and vertical checkpoints. We also investigated the influence of surface roughness on the accuracy of DEMs and resultant DoDs.

The experiment was set up outdoors on a level table (1.2 m x 0.6 m). Two scaled coded markers (0.25 m) and a series of 2.5 cm x 2.5 cm checkboard pattern, non-coded markers and eight, evenly spaced wooden blocks of known dimensions were fixed onto the table surface (Figure 1 C). These were used to calculate the horizontal and vertical error of the DEMs (as recommended by Verma and Bourke). Four simulated platform surface blocks were constructed using moulds, gypsum plaster. Stainless steel, square-headed bolts for mounting the CRS, as described above, were installed on each block. A digital inclinometer (Examobile Bubble Level for iPhone) was used to ensure the surface of the bolt was level. The surface of the experimental blocks were constructed to represent a range of micro to meso scale roughness that are observed in the field. These include low (B1), medium (B2) and high (B2) relative surface roughness (Figure 1 D-F). All blocks were sprayed with matte grey paint to allow easy identification of 'erosion' areas and provide additional visual validation of the models. A set of three 1 cm x 1 cm checkboard non-coded markers were fixed to each experimental block to serve as additional checkpoints for horizontal error. One block (B-con) was used as a control. The remaining three blocks (B1, B2 and B3) were used to carry out the experiment. Each block was placed at the centre of the table when acquiring images.





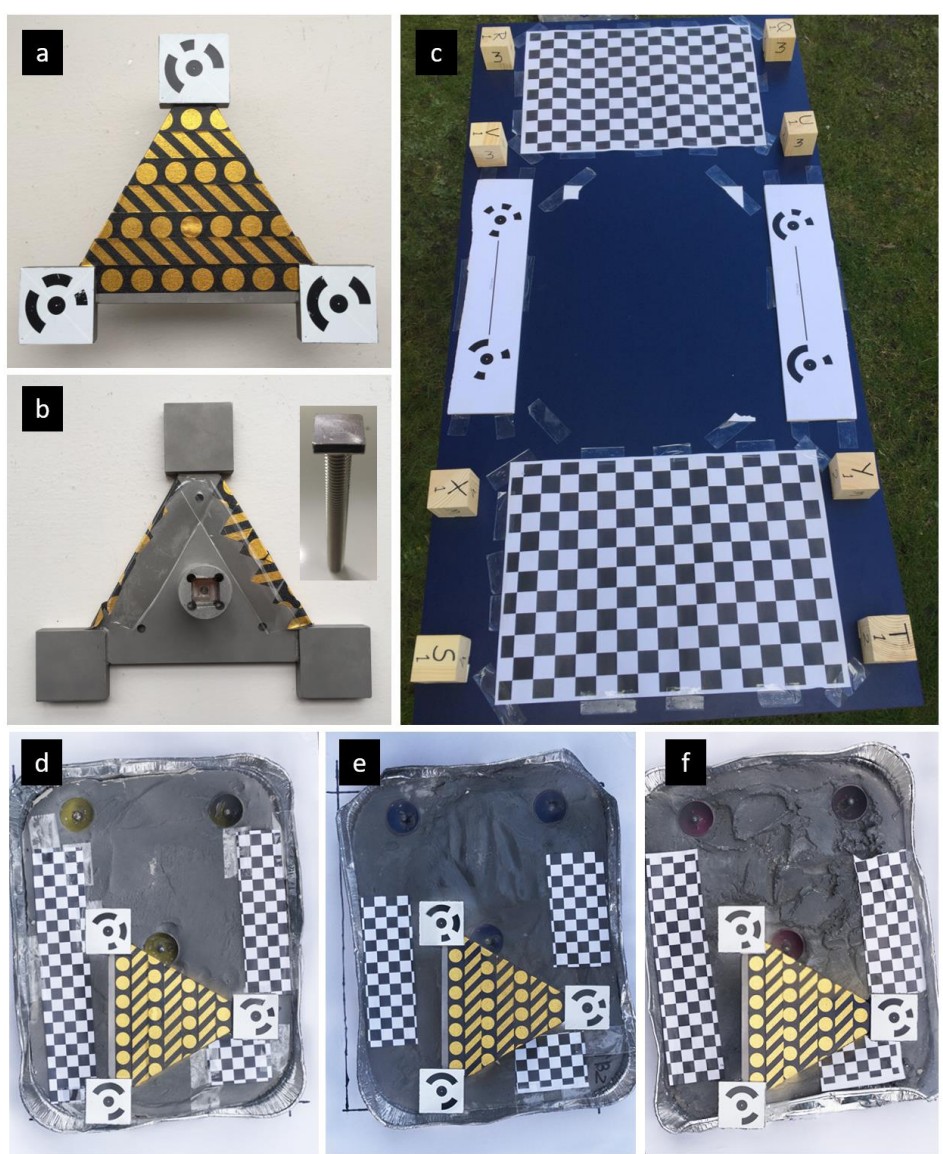

**Figure 1.** The experimental setup. a) The CRS top view and b) underside with the square headed bolt (inset) c)
The experimental platform with markers and wooden blocks used to calculate the horizontal and vertical error. D,
e, f) The simulated platform surface of experimental blocks B1, B2 and B3 designed with increasing roughness

### 2.3 Data collection

In order to replicate field conditions as closely as possible, all images of the experimental blocks were acquired
outdoors during a single day. The CRS was placed on the pre-installed square head bolt, and orientation was noted.
We used a Nikon D5500 with a variable zoom lens set up at 24 mm focal length.. Approximately 100 images of





each block were obtained. This number of images was required to capture the full extent of the table with the non-coded markers and the wooden blocks used for the error analysis. We expect that 40-50 images would be sufficuient to generate a high-resolution DEM for a smaller area (e.g. <0.5 m$^2$) in the field. In this study, ~70 images were acquired at a distance of ~1m from the experimental blocks and then a series (25-30 images) of close-range shots at <0.5 m. All three experimental blocks and the control block were imaged on the table prior to

simulating erosion on the blocks.

Recent work has demonstrated the potential efficacy of smaller-scale physical erosion processes on high energy Atlantic rock platforms (Cullen and Bourke, 2018). However accurate quantification of these features has not been possible. We therefore tested simulations of three known types of erosion: 1. Sweeping abrasion was simulated by gently abrading the surface of all three blocks with a medium grit sandpaper. 2. Scratches were

simulated using a screwdriver     3. Impact percussion marks were simulated on one block using a hammer and chisel.

The CRS was removed and replaced between each stage of data collection, as would be practice for carrying out repeat surveys in the field. Images of the blocks were taken following simulated erosion as outlined above.

**2.4 Repeatability**

The utility of this approach for microscale change detection using the CRS developed for this study is contingent on the exact replacement of the CRS during successive surveys in the field. To test the repeatability of this approach, we used a control block to acquire images for DEM generation using the data collection and processing procedure outlined above. At the end of the experiment, the CRS was replaced and a second series of images were acquired for DEM generation for comparison. DEM accuracy and error propagation were calculated as described

below.

**2.5 Data processing**

**2.5.1 Digital Elevation Models**

All the images were acquired in raw format during the experiment. RAW images were converted to 14-bit

uncompressed tiff format with AdobeRGB colour space in Adobe Lightroom. We used Agisoft Photoscan (version 1.4.1). Image quality (Q) was assessed using the Estimate Image Quality tool in AgiSoft and images with Q values < 0.5 were removed. The CRS was used to scale and georeference the model. Baseline DEMs and orthophoto mosaics for each block were generated and exported at the highest, common pixel resolution (0.3mm/pixel).

**2.5.2 DEMs of Difference**

DEMs were exported in ArcMap, and a polygon shapefile was drawn over the area of interest for each block. The area of interest i.e. the erosion area of the simulated platform surface, was extracted for analysis using the Extract by Mask' tool in Spatial Analyst tools. DoDs were generated using the Raster was Math tool (minus) in ArcMap (version 10.5) using Eq. (1),





$$B1 \, DoD_1 = B1 \, DEM_1 - B1 \, DEM_0 \qquad (1)$$

where the subscript refers to the experimental stage.

### 2.5.3 Rugosity

To permit evaluation of the impact of different degrees of surface roughness on the accuracy and reliability of our
generated DEMs, a rugosity index for each block was calculated in ArcMap using the standard Surface Area ratio
method (Dahl, 1973; Risk, 1972) where,

$$Rugosity = \frac{Contoured \ area}{Planar \ area} \qquad (2)$$

A rugosity index (RI) of 1 indicates a planar surface while increasingly higher values indicate increasingly
'rougher' surfaces. The contoured area for each block was calculated using the relevant baseline DEM. A TIN
surface was generated using the Raster to TIN tool in ArcMap. The contoured surface area for the specified region
was calculated using the Polygon Volume tool in ArcMap. The planar surface area of the same region was derived
using the calculate geometry tool assuming negligible change in slope over the specified area. The RI was
calculated using Eq. (2).


### 2.5.4 DEM accuracy and error propagation

The coded and non-coded markers fixed to the table were used as checkpoints to determine the horizontal (XY)
error of the DEMs produced using the CRS (Verma and Bourke). For each DEM, the model and its respective
orthophoto were imported into ArcMap (version 10.5) and the distance between 30 randomly selected checkpoints
and the two coded scale bars (Figure 1) were measured using the measurement tool. The horizontal error was
calculated as the Root Mean Square Error (RMSE) of the difference between measured length and known length.

To determine the vertical accuracy of the DEMs, eight wooden blocks were used as checkpoints (Figure 1). The
DEMs and orthophotos were imported in ArcMap where the height of wooden blocks were measured using the
Interpolate Line tool, by drawing a line across one of the sides of the wooden block and extending it to the ground
surface. We ensured that the line drawn was straight. Height was estimated as the difference in mean elevation
between wooden block top surface and the surrounding ground surface on each side. The actual height of wooden
blocks was measured by an electronic digital Vernier Caliper. The Vernier Caliper has an accuracy of 0.03 mm
and measurement repeatability of 0.01 mm. We obtained five measurements along the same side of wooden block
measured in ArcMap. We used the mean of these five measurements to calculate the height of the wooden block.
The actual height was subtracted from the estimated DEM height to calculate the vertical error.

The calculation of a DoD can result in propagation of error associated with the DEMs used in the computation
process. As such, an error analysis is required to increase confidence in the DoD results. This is particularly



important when the scale of geomorphic change being detected is of similar magnitude to uncertainties of the DEMs used in the DoD calculation.

We determined the minimum level of detection as the most suitable method of error analysis for this study as the development of shore platforms is primarily an erosional process and as such, the spatial coherence of erosion and deposition (Wheaton et al., 2010) is unsuitable as a method for error analysis in this study. Additionally, while probabilistic approaches produce reliable estimates of morphological change (e.g. Brasington et al., 2003; Brasington et al., 2000; Lane et al., 2003), small changes in elevation, such as those measured in this experiment,

may be disguised as noise (Williams, 2012).The minimum Level of Detection (LoD) uses the quadratic composition of errors in the original DEMs to estimate the propagated error of the calculated DoD (Brasington et al., 2003; Gómez-Gutiérrez et al., 2014; Lane et al., 2003; Wheaton et al., 2010; Williams, 2012):

$$E_{DoD_{1-2}} = \sqrt{(E_{DEM_1}{}^2 + E_{DEM_2}{}^2)} \tag{3}$$

Where $E_{DoD_{1-2}}$ refers to the LoD calculated as the square root of the combined squared errors of the DEMs used to

generate the DoD. If values of $E_{DEM_1}$ and $E_{DEM_2}$ are known, this method can be applied at a global or local scale where the spatial variability of the error terms are known (Lane et al., 2003). We applied Eq. (3) to determine the minimum threshold of detection for each DEM (Williams, 2012) for each stage of the experiment. Changes detected that fall within the limits of detection (+ $LOD_{min}$ or – $LOD_{min}$) calculated using Eq. (1) are considered noise and interpreted as no change.


## 3 Results

### 3.1 Accuracy and error propagation

DEM generation resulted in a maximum and minimum horizontal (XY) RMSE of 0.23 mm and 0.03 mm respectively. Maximum vertical (Z) RMSE was 0.52 mm with a minimum of 0.23 mm. The minimum limit of

detection was calculated at 0 ± 0.27 mm while the maximum LoD was 0 ± 0.71 mm.

**Table 1.** The horizontal (XY) and vertical (Z) RMSE error for the control block (B-con) and the experimental blocks B1, B2 and B3. LoD for each DoD is also shown.

| DEM | XY RMSE (mm) | Z RMSE (mm) | LoD (0 ± mm) |
|---|---|---|---|
| **B-con** | | | |
| 1 | 0.03 | 0.45 | N/A |
| 2 | 0.12 | 0.23 | 0.27 |
| **B1** | | | |
| Stage 0 | 0.23 | 0.37 | N/A |
| Stage 1 | 0.12 | 0.39 | 0.54 |
| Stage 2 | 0.22 | 0.44 | 0.56 |




| | | | |
|---|---|---|---|
| Stage 3 | 0.12 | 0.52 | 0.71 |
| **B2** | | | |
| Stage 0 | 0.1 | 0.40 | N/A |
| Stage 1 | 0.2 | 0.46 | 0.53 |
| Stage 2 | 0.1 | 0.35 | 0.49 |
| Stage 3 | 0.2 | 0.45 | 0.56 |
| **B2** | | | |
| Stage 0 | 0.2 | 0.39 | N/A |
| Stage 1 | 0.1 | 0.37 | 0.54 |
| Stage 2 | 0.1 | 0.39 | 0.54 |
| Stage 3 | 0.1 | 0.45 | 0.60 |


### 3.2 Repeatability

The change in vertical elevation for the control block calculated from the DoD is shown in Figure 2 below. The maximum change in elevation ( - 0.29 mm) is within the LoD and is interpreted as no change.

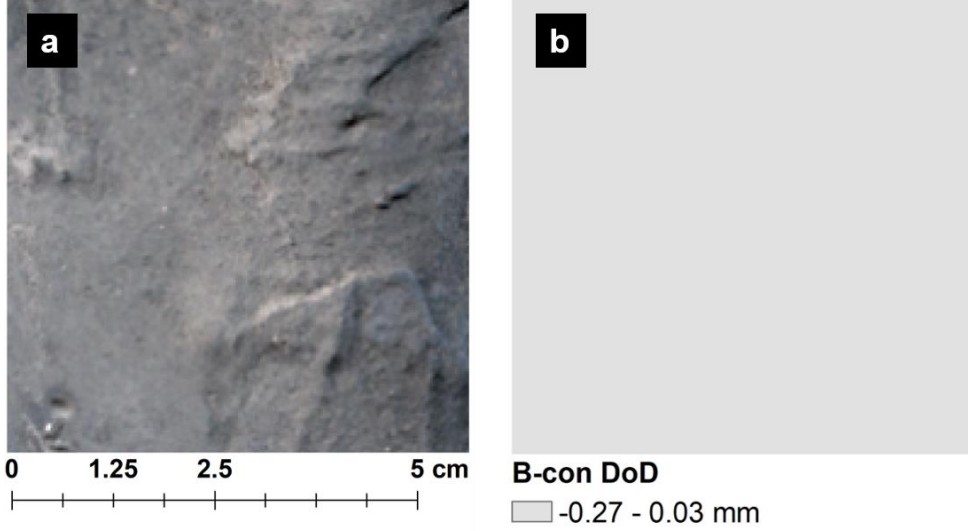

**Figure 2.** (a) The control block (B-Con) orthophoto and (b) DoD showing a change in surface elevation between successive DEMs.





### 3.3 Rugosity

The RI calculated for each block is shown in Table 2. The control block (B-con) had the lowest rugosity (planar

surface) while B1 had a very low RI followed by B2 and B3 in order of increasing rugosity.

**Table 2**. Contoured surface area (SA), planar surface area and Rugosity Index (RI) for each of the experimental blocks.

| Block ID | Contoured SA (mm) | Planar SA (mm) | R Index |
|----------|-------------------|----------------|---------|
| B-con | 8.9 | 8.9 | 1.00 |
| B1 | 9.0 | 8.9 | 1.01 |
| B2 | 11.7 | 10.9 | 1.07 |
| B3 | 9.9 | 8.2 | 1.21 |

**3.3.1 Very low rugosity platform: B1**

The results for experimental block B1 are shown in Figure 3 (a-i). The surface area of B1 used in the analysis is shown in (a) where light grey indicates the area of abrasion. For B1 Abrasion, a maximum negative surface change of 1.06 mm was detected, while an increase of 0.30 mm was observed (b) before the LoD was applied. The area of negative surface change between 0.1 mm and 1.06 mm corresponds to the actual area abraded. After

thresholding at the LoD, the area of change detected is significantly lower (less than half) the area where the actual change occurred. No increase in surface elevation was detected. For B1 Scratches, the scratched surface is shown in d (black arrows). Before thresholding, the maximum negative change on the surface of B2 was 0.35 mm while an increase in surface elevation of 0.26 mm was detected. Negative changes corresponded well to the observed locations of scratches. After thresholding at the LoD, no changes were detected on the block surface (f). For B1

impact percussions, the locations where block fragments were removed are shown in G (black arrows). Maximum negative change detected, i.e. predicted depth of percussions, was 1.49 mm, while a positive change in surface elevation of 0.30 mm was detected before thresholding (H). After thresholding, no positive change in surface elevation was detected and predicted negative change corresponded well to the actual location of  percussions (i).

To summarise, for a simulated platform with a very low RI, sweeping abrasion and chips were reliably detected

in the thresholded DoD. Scratch depths were less than the LoD  and as such were not detected in the thresholded model.



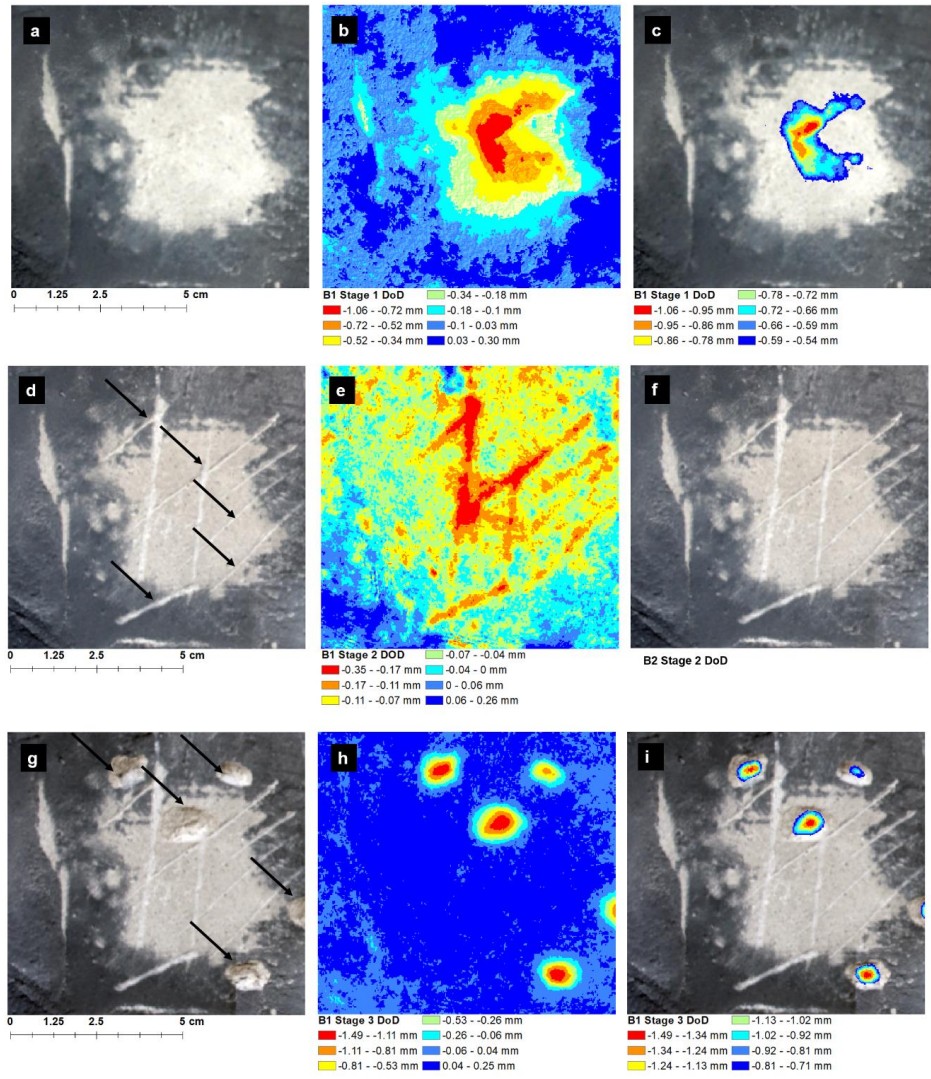

**Figure 3.** (a) B1 Stage 1 Orthophoto showing abraded surface of simulated platform surface (light grey). (b DoD for B1 Stage 1 before thresholding at LoD and the thresholded DoD (c. (d) B1 Stage 2 orthophoto showing location of scratches, (e) B1 Stage 2 DoD before thresholding and (f) DoD shown in E thresholded at LoD. (g) B1 Stage 3 orthophoto showing locations of percussions. (h) B1 Stage 2 DoD before thresholding at the LoD and the thresholded DoD (I).



### 3.3.2 Moderate rugosity platform (B2)

The results for experimental block B2 are shown in Figure 4 (a-i). Abraded surface area is indicated by lighter toneareas in Figure 4a.   While this abrasion is visible in the DOD (Figure 4b), a significant component of the detected change occurred where no change was expected. This corresponds to 'shadow zones' associated with topographic highs. This resuly ws not affected by thresholding at the LoD (Fig. 4c).

Scratches are evident in Figure d.  Furthermore, the location of negative change corresponds well to the location

of scratches (Fig. 4e). However, similar to B1,  a small area of change is detected around the deepest scratch where none is expected (Fig. 4f). The impact percussion features are shown in (Fig. 4g). The maximum negative change in the surface elevation detected ( i.e. the depth of percussions), was 3.35 mm, while the maximum positive change was  0.57 mm (h). Following  thresholding, no positive change in elevation was detected (Fig. 4i) and negative change corresponded well to the actual location of percussions.






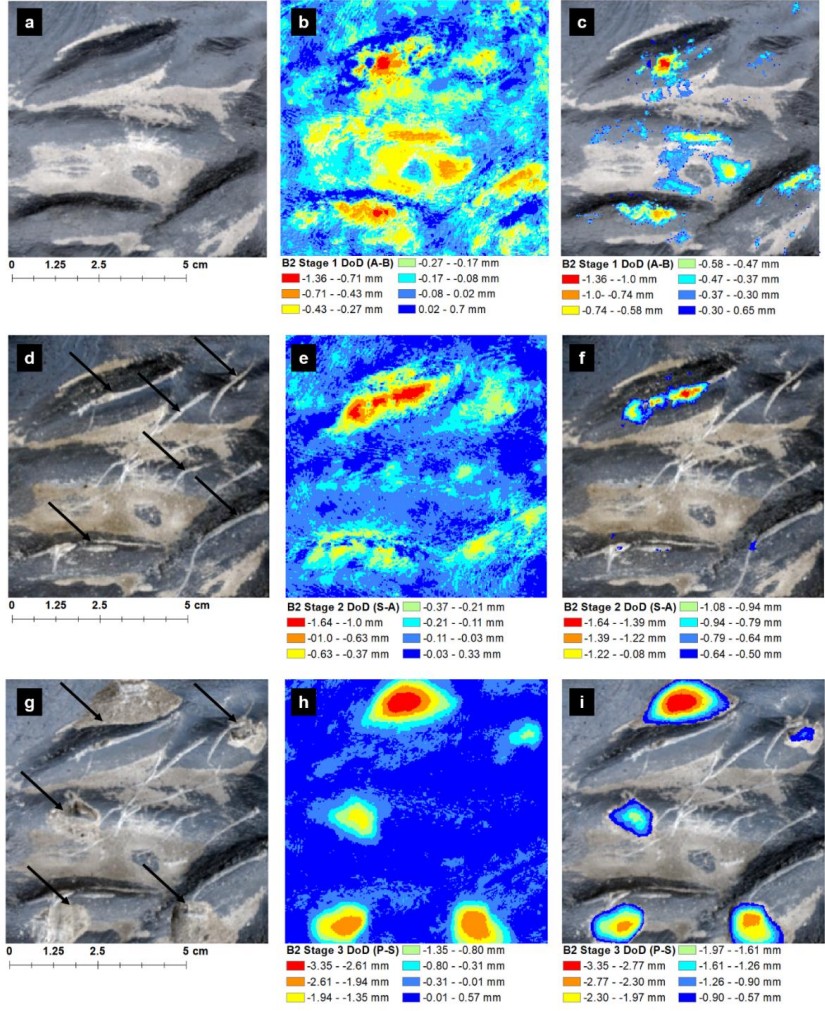

**Figure 4.** (A) B2 Stage 1 Orthophoto showing abraded area (light grey) of simulated platform surface, (B) B2 Stage 1 DoD before thresholding at LoD and the thresholded

DoD (C). (D) B2 Stage 2 orthophoto showing scratched surface of B2 (black arrows). (E) B2 Stage 2 DoD before
thresholding and (F) DoD shown in E thresholded at LoD. Note change detected in shadow zones in F (white arrow) where none is expected. (G) B2 Stage 3 orthophoto percussed surface. (H) DoD before thresholding at LoD and (I) DoD thresholded using calculated LoD.

To summarise, for a simulated platform with a moderate RI, only scratches and impacts were detected in the thresholded DoD.



### 3.2.3 Relatively high rugosity paltform (B3)

The results for B3 are shown in Figure 5 (a-i). The light toned areas in (a) indicate the abraded surface of the experimental block. In general, the maximum negative change detected (red and orange areas in b) correspond well to the abraded area. However there are significant increases and decreases (>3 mm) in surface elevation

where no change was expected. As above, the largest of these errors generally occurred in 'shadow zones'. Thresholding did not significantly improve the resultant DoD (i). For scratches (Fig. 5d)there was a reduction in surface elevation of 3.45 mm detected where no change was expected. As with the previous stage, these changes were observed to occur in shadow zones. Thresholding at the LoD did not improve the resultant DoD, and both increases and decreases were recorded where no change was expected (white arrows in i). The location percussions

are shown in Figure 5g. Maximum negative change detected corresponded mainly to the location of percussion however negative change was recorded where none was expected (h). As before, abnormal change occurred in shadow zones. Thresholding improved the resultant DoD (i), and the majority of negative change observed corresponded well to the location of percussions, except in some small areas (white arrows in i), associated with shadow zones. Maximum percussion depth was recorded at 5.43 mm.

To summarise, for a simulated platform with a relatively high RI, only impacts were reliably detected in the thresholded DoD. However there were errors (larger than in B2) in the data, which are concentrated in topographic 'shadows'.



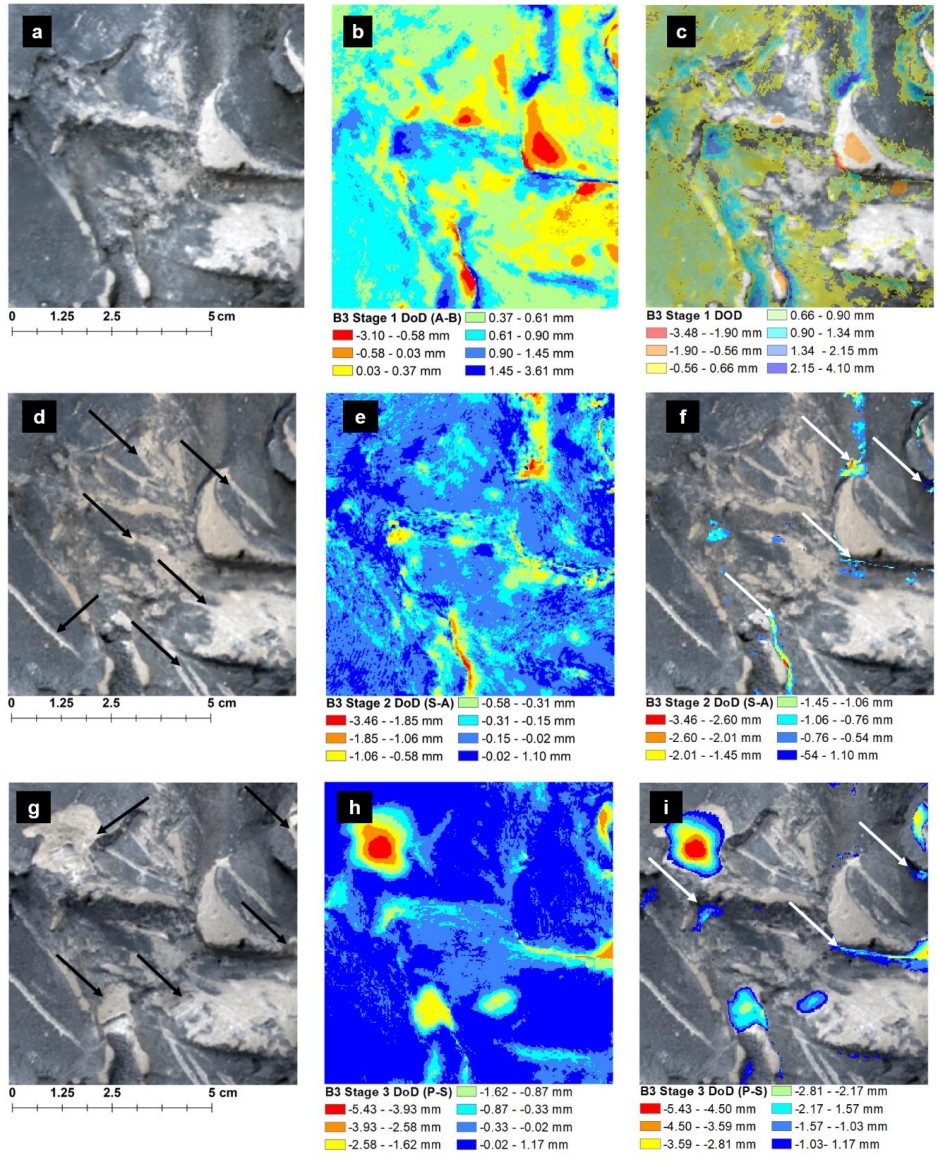

**Figure 5.** (A) B3 Stage 1 Orthophoto showing abraded surface area (light grey) of simulated platform surface. (B) DoD for B3 Stage 3 before thresholding and (C) DoD at LoD shown at 50% transparency overlain onto the orthophoto shown in A. Note significant geomorphic change detected in shadow zones (white arrows) where no change is expected. (D) B3 Stage 2 orthophoto showing scratched the surface of the simulated platform (black arrows), (E) DoD before thresholding and (F) DoD thresholded at LoD. As in C, note change higher than the LoD detected in shadow zones (white arrows) in F where no change is expected. (G) B3 Stage 3 orthophoto showing



the location of percussions (black arrows), (H) B3 Stage 3 DoD before thresholding at LoD and (I) B3 Stage 3 DoD thresholded using calculated LoD. Note shadow zones (white arrows) where DoD indicates change, but none is expected.

### 3.3 Comparison of the T/MEM and SfM-MVS for measuring erosion on rock shore platforms


The T/MEM has, over decades, cemented its position as a low-cost method for measuring microscale erosion on shore platforms, while SfM-MVS is fast emerging as a valuable tool in the geomorphologists toolkit for the detection and measurment of geomorphic change at a range of scales. Both approaches have advantages and limitations and the choice for use one method over another will depend on a number of factors such as cost, the ease of data collection and the quality and value of the data obtained.


We have compared our experience of using the T/MEM to that of the SfM-MVS (based on the CRS and workflow used in this study) as a means for detecting geomorphic change on rock shore platforms under the following headings. We evaluated both techniques for; ease of data acquisition (including both installation and data collection), data processing, hardware costs, software costs, model resolution, accuracy and overall ease of use. Our reported installation, data collection and data processing times refer to a single measurement station. Hardware costs for the TMEM are based on initial outlay for SDS drill, drill bits, the TMEM platform and engineers gauge. Hardware for the SfM-MVS workflow described in this study refers to initial outlay for the manufacture of CRS and cost of the camera. Basic hardware costs (e.g. computer for processing) are not included. Overall ease of use for each method is based on our experience of data acquisition in the field (installation and collection) and data processing. An overall comparison is provided based on the above factors in addition to the value of the data obtained.



A comparison of the TMEM and the SfM-MVS approach as a means for detecting geomorphic change on rock shore platforms is shown in Figure 6. Both methods have clear advantages and disadvantages and the comparison is intended to be a guide to assist researchers in choosing the most appropriate method for specific project deliverables.


#### 3.3.1 Installation

To install a single T/MEM measurement station, three holes are drilled at the apex of an equilateral triangle and pins set into each hole with a marine grade epoxy resin. The time needed to install a single TMEM station varies between 40 and 80 minutes depending on rock hardness. For the workflow used in this study, the time needed to install a single bolt to mount the CRS will take approximately one-third of the time.






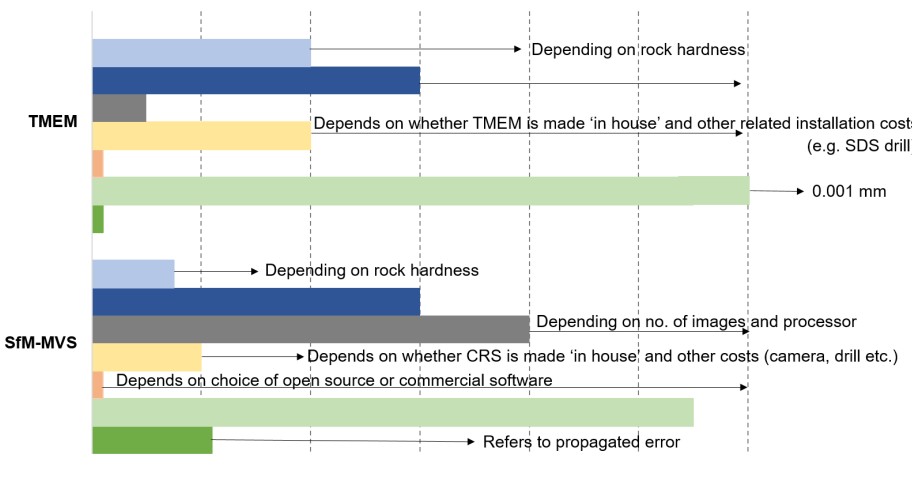

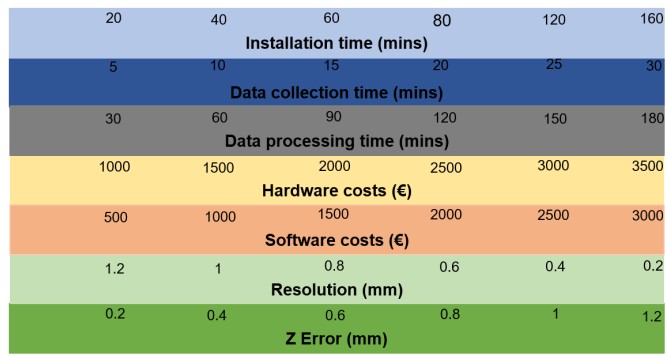

**Figure 6.** Comparison of TMEM and the SfM – MVS workflow presented in this study under different categories. Values showed (cost, time etc) increase from left to right apart from 'Resolution' where decreasing values from left to right indicate increasing resolution.


### 3.3.2 Data collection

In our experience, the time needed to collect data from a single station (based on 100 measurements) using a TMEM varies between 15-30 mins (grey bar in figure 6). This will depend on whether the digital gauge being used has a USB memory, which automatically stores measurements as they are taken (e.g. Stephenson, 1997), or

whether measurements are recorded manually which increases the time required. In comparison, acquiring the 40-50 images required for SfM-MVS took approximately 15 minutes.



### 3.3.3 Data processing

The time required to process TMEM data will depend on the number of measurements collected and the method used to record data in the field, i.e. whether they are stored automatically (e.g. Stephenson, 1997) or manually.

Automatic recording reduces the time needed to process data however manual processing can take up to 30 minutes per station (based on 100 measurements). Data processing takes significantly longer for SfM-MVS (2-3 hours per DEM) depending on number of images and the processor used.

### 3.3.4 Hardware costs

The cost of a TMEM platform varies considerably depending on whether it is made in-house or commercially. In-house construction is considerably less (~€900 for materials and labour), while a commercial TMEM costs approximately €2000 (based on 2017 prices). The cost of the digital gauge also varies depending on the manufacturer, model, resolution, accuracy and Ingress Protection (IP) needed and range from €200-€500. Most rock types will also require an SDS drill with masonry bits which cost in the region of €600. The cost of the 316

stainless steel pins also varies depending on whether they are constructed 'in-house' or purchased commercially.

### 3.3.5 Software costs

Software cost for TMEM data processing is negligible while there are free open source software available for processing of images for SfM–MVS (e.g. VisualSfM). However, commercial packages such as Agisoft Photoscan

can cost between €600 and €3500 depending on the licence type (e.g. Pro, Standard, Educational, Stand alone or Floating).

### 3.3.6 Resolution and Error

Depending on the digital gauge used, TMEM measurements can have a resolution of up to 0.001 mm with a

reported measurement error of ± 0.005 mm (Gómez-Pujol et al., 2007). Resolution for SfM-MVS (achieved in this study) was 0.3 mm per pixel. For some DEMs it was less than this (0.15 mm per pixel) however differencing of DEMs requires that pixel resolution be the same for both DEMs being compared. The CRS and SfM-MVS workflow employed for this study achieved maximum XY and Z RMSE errors of 0.2 mm and 0.5 mm respectively.

**4 Discussion**

The T/MEM has contributed significantly to our understanding of microscale erosion processes on shore platforms. Measurements of microscale platform erosion using a T/MEM are limited to repeated point measurements over time which provide a mean rate of surface downwearing within the measurement area for that measurement period with the dominant process(es) being inferred from the spatial and temporal variation in

downwearing rates (Trenhaile, 2003). However, the method's inability to measure erosion at different scales was noted by Stephenson and Finlayson (2009) as a limitation and the authors advocated the introduction of new




methods for measuring shore platform erosion at a range of scales. We have developed a CRS which can be quickly deployed by researchers in the field for detection of micro and meso-scale erosion on shore platforms using SfM-MVS Photogrammetry and a geomorphic change detection approach. The CRS described in this study

permits rigorous georeferencing of DEMs derived using the SfM-MVS workflow.

We have demonstrated that SfM-MVS Photogrammetry can be used to reliably detect sub-mm changes on shore platforms where the platform surface has a low RI. This approach successfully detected 0.3 mm downwearing of the simulated platform surface of B1 caused by abrasion of the surface. While we were also able to identify shallow scratches on surface of the experimental block, applying the LoD obscured this finding due to the shallow

depth of scratches (< 0.3 mm). However, we were able to detect loss of mm-cm sized rock effectively. This demonstrates that our approach offers a method for cross scalar analysis of erosion on shore platforms, offering a much-needed means to examine relationships between micro and meso-scale processes of shore platform erosion and morphologies.

Our results indicate that as RI increases, the reliability of SfM-MVS for detection fine scale erosion is reduced

due to increased topographic complexity. Despite areas of reduced elevation, i.e. erosion, aligning well with areas where the surface had been abraded, there were areas of change where clearly none was expected. Despite this, our approach successfully detected the loss of rock fragments on the simulated platform surface of B2 (higher RI) once the LoD was applied. Similarly, for B3, which had the highest RI, fine-scale erosion and scratches were not detected reliably, and while the loss of rock fragments was detected, the effect of complex topography in creating

shadows zones produced abnormal change. The orthophotos were important in this regard as they provided visual validation of the models and highlighted the influence of shadow zones in introducing error into the models. The additional uncertainty introduced into the models due to the surface complexity was not accounted for using the LoD approach. This resulted in abnormal change detection associated with meso scale ( > 1mm) slopes and troughs. While the strong influence of surface complexity may be considered a limitation, it should be noted that

the T/MEM is largely restricted to measurements of downwearing on small surface areas with low topographic complexity. As such, it does not exclude this approach as an alternative for measuring change on this type of surface.

Precision mapping (James et al., 2017) offers a potential approach to address this as there is an opportunity to increase confidence in the accuracy of point clouds derived for more complex platform morphologies. While the

LoD assumes a global uniform distribution of error, precision mapping explicitly accounts for the spatially variable precision characteristic of photo-based surveys (James et al., 2017) and has been demonstrated to improve change detection in areas with complex topography. Future work will test this approach.

This study and our experience in the field using a TMEM suggest that the time required for data collection (installation and acquisition) collection is shorter using an SfM-MVS approach compared to the TMEM. The

requirement of just one bolt per measurement site for the CRS described here, compared to three bolts per measurement site for the TMEM, reduces the time needed for initial installation in the field. Add to that the time required to collect images for the SfM-MVS workflow compared to the time required to collect 100 TMEM measurements, and SfM–MVS has notable advantages. This reduced installation and data acquisition time are of particular worth for shore platforms with meso to macro tidal ranges, where time in the intertidal zone is limited



to, at most, a couple of hours either side of low tide. For larger platforms, where a number of measurement stations are located in the intertidal zone, time is clearly a limiting factor, and methods which alllow rapid installation data collection are preferable.  Regarding data processing, the time required depends on the gauge used to collect the TMEM data, i.e. manual or automatic and the desired output (point measurements or 3D surface).  Regardless, the processing time required for SfM-MVS is significantly higher (2-3 hours per DEM generated). Nevertheless,

batch processing options in Photoscan mean that DEM generation process/steps can be automated and the user time on computer is reduced. With respect to image acquisition for SfM-MVS, we used a Nikon D5500 and had included this in our overall analysis however expensive cameras are not a prerequisite. For example, in a recent experimental study of surface features in sand caused by sublimation of $CO_2$ ice, of a similar scale to this study, Mc Keown et al. (2017) used an iPhone to acquire images and utilised the same CRS developed in Verma and

Bourke ( to scale and reference DEMs, achieving similar accuracy and resolution (<1 mm).

The T/MEM offers considerable resolution and accuracy for measurements of very small surface changes, which is particularly useful for measuring very slow rates of downwearing and detection of very small changes due to platform swelling (e.g. Gómez-Pujol et al., 2007; Hemmingsen et al., 2007; Porter and Trenhaile, 2007; Stephenson and Kirk, 2001; Trenhaile, 2006). For faster-eroding rocks, the precision obtainable using a  T/MEM

is not required (Stephenson and Finlayson, 2009). While the highest common resolution of the DEMs produced for this study were 0.3 mm/pixel, this is demonstrated to be sufficient for measuring micro-scale and meso scale erosion on surfaces with low RI and loss of rock fragments on more topographically complex surfaces.

In terms of data output, the TMEM produces a series of surface point measurements. These can be compared directly to point measurements made from previous surveys or plotted as a digital elevation model for 3D

visualisation of the surface at the bolt site (e.g. Stephenson, 1997). The spatial and temporal variation in downwearing rates can be used to infer the efficacy of erosion processes. In this, we suggest that SfM–MVS has a clear and important geomorphic advantage. The technique produces point clouds and DEMs which can be used to identify and classify surface features as well as detect geomorphic change at different scales. This added value in the approach is significant. Orthophotograph mosaics offer additional means for validating meso scale changes

on the rock surface and identifying erosion styles.

## 5 Conclusions

1. This study demonstrates that SfM can be used to detect sub-mm changes due to erosion on shore platforms. However, we find that as the complexity of the rock surface topography increases, the reliability of SfM to detect

sub-mm changes decreases. We note that the application of TMEM is also limited to relatively planar surfaces. Future work will test the precision mapping approach of James et al. (2017) to determine spatial distribution of error and increase confidence in results on more topographically complex platform surfaces.

2. While TMEM has higher resolution and accuracy compared to SfM, if offers a limited number of point measurments over a small area. In comparison, SfM produces 3D topographic data from dense point clouds and

DEMs which can be used to identify, classify and quantify different styles and scales of erosion.

3. In this study, we have provided a detailed comparison between TMEM and SfM methods to measure change due to erosion on rock surfaces in the coastal environment.

**Data availability**

All data is available upon request from the corresponding author at cullenni@tcd.ie

**Author contributions**

N.D.C. prepared and wrote the main body of the manuscript with discussion and contributions from A.K.V. and M.C.B. N.D.C. designed the experiment with input from A.K.V. and M.C.B. A.K.V. designed the original

coordinate reference system and N.D.C designed the field adapted version with input from A.K.V and M.C.B. N.D.C and A.K.V carried out the experiments. A.K.V. processed images and generated the digital elevation models. N.D.C. carried out data analysis. A.K.V. and M.C.B reviewed and edited the final manuscript.

**Acknowledgements**

N.D.C and A.K.V were supported by the Trinity College Dublin Postgraduate Studentship, Faculty of Engineering, Mathematics and Science. The authors would also like to thank the Geography Department at Trinity College Dublin for additional support in carrying out this research. Our thanks also to Neil Dawson from J.F. Dunne Engineering for expertise in the manufacture of the field adapted coordinate reference system used in the experiment.

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
