# Peer review of "A comparison of Structure from Motion Photogrammetry and the Traversing Micro Erosion"

_Earth Surface Dynamics, 2018_

## Referee Comment (RC1) · W. J. Stephenson (Referee) · 16 Jul 2018

esurf-2018-55 Submitted on 02 Jul 2018 A comparison of Structure from Motion Photogrammetry and the Traversing Micro Erosion Meter for measuring erosion on rock shore platforms Niamh D. Cullen, Ankit K. Verma, and Mary C. Bourke

This paper offers an interesting and useful comparison between SfM and the TMEM methods. It helps to demonstrate how erosion measurements on shore platforms can be made across a range of scales from sub-millimetre to centimetres. While this is

welcome I think the absence of a genuine field trial and data from a shore platform is a significant weakness. I suspect that such data are available and hope that some could be presented in a revised version of the manuscript to show that it actually worked in the field. This might help address my second main concern related to rock swelling. Throughout the paper there are many places where further reference to rock swelling is needed. Certainly in the consideration of SfM versus TMEM the value of the TMEM for the study of rock swelling has not been given enough attention. In addition to swelling the TMEM has enabled much shorter timescales to be considered. At line 53 – there needs to be a correction, since short times scales are actually hours not years based on a number of studies that have investigated rock swelling over hours and days. Can your SfM method detect short term variability in rock surfaces? Reference to this point is needed at line 115 needed.

Where multiple references are provided these must be in chronological order, not alphabetical (I suspect that is an artefact of using reference management software). It is important that chronological order is used so to recognised correct attribution and who made the contribution first. E.G. it is important that Spate et al is recognised before Moses et al 2014, since Spate clearly identified sources of error long before Moses.

Scale terms need far more careful definition, micro – meso and macro scales need to be defined with a range of values. E.g. at line 165, what is micro to meso?

I wondered if it is possible to expand the applicability of the paper by reference to rock erosion more broadly than just shore platforms. This should probably include reference to: Turowski, J.M. and Cook, K.L., 2017. Field techniques for measuring bedrock erosion and denudation. Earth Surface Processes and Landforms, 42(1), pp.109-127. After all the method is not limited to shore platforms and clearly has potential in any environment where bedrock erosion is of interest.

I find the use of the term "rock shore platform" unnecessary, it is almost a tautology – just shore platform.

Line 65 The limitation to smooth surfaces is pretty well known, I don't think you can claim to have identified this limitation.

Line 142, what does "specially" mean? Needs explanations.

Line 145 how accurate is accurately? +/- how many millimetres?

Line 148 how was the bolt levelled? Move explanation from lower down up.

Line 150 you need to tell the reader what is the "high degree of relocation precision". Is this same as the Kelvin clamp used by the TMEM? Or is it +/- some number and unit?

Since this is a technical methods paper I think you need to provide a technical drawing of your bolt and plate so others can manufacture both themselves.

Fragments and granular need definition, what size are these?

Caption to Fig. 1 can you add some dimensions of the triangle angles so we get a better sense of scale?

Was the camera held by hand or placed on a tripod? Does it matter? Line 186 again what does "smaller-scale" mean?

A key point you have not made and this becomes evident at start of the results is that SfM is an order of magnitude lower in precession than the MEM.

Lines 368-69, actually I think the key is not factors such as cost (they are actually pretty similar if you cost the camera) but what questions are being asked and so at what scale are measurements required.

Installation times are way off here, it does not take 80 minutes to install a MEM site. An experienced operator with a good drill, can install a site in 20 minutes or less. If two people are operating together it can be very fast. The other factor is the rock type, more resistant rock, then yes a bit slower.

In section 3.3 you need to discuss the benefit the TMEM provides for investigating rock

swelling, something you have not really dealt with, and not (yet) shown to be detectable by the SfM method. You need to say something about this in your discussion section as well.

Minor typos Line 63 Stephenson and Kirk 1966 – probably 1996 or is it 1998?? Cullen and Burke 2018 not in reference list. Verma and Burke without year or in press. Line 62 use and between moulds and gypsum

---

## Short Comment (SC1) · 20 Jul 2018

Dear Dr. Stephenson,

We thank you for your review of the manuscript entitled 'A comparison of Structure from Motion Photogrammetry and the Traversing Micro Erosion Meter for measuring erosion on rock shore platforms'.

We appreciate your feedback and will address your comments in due course.

Many thanks,

Niamh Cullen.

---

## Referee Comment (RC2) · Anonymous Referee #2 · 14 Aug 2018

The article "A comparison of Structure from Motion Photogrammetry and the Traversing Micro Erosion Meter for measuring erosion on rock shore platforms" compare a methodological implementation of SfM-MVS photogrammetry to evaluate sub-millimetre erosion processes of shore platform, with classical direct measurements from TMEM system. The structure of this paper is composed of: 1) an introductory part resuming the methodologies used to measure gentle topographic changes over rock shore surface with focus on TMEM and SfM-MVS photogrammetry; 2) a detailed methodological section; 3) the results, segmented between a quality assessment of

DEMs, a comparison of the level of geomorphological changes detection regarding the rugosity of surface, and a comparison between SfM-MVS photogrammetry tools and TMEM; 4) a discussion and conclusion.

My opinion about this article is that it proposes a very interesting approach to quantify and map small scale erosion processes over shore platform at low cost and with accuracy. The SFM-MVS photogrammetry protocol and results are very detailed, especially regarding vertical error assessment. The results show that the method is able to reach small and medium scales detection of erosion over low rugosity surface but are clearly limited to detect loss of rock fragment over more complex topography. The SfM-MVS photogrammetry results show typical errors of geometry reconstruction associated to the technique such as shadow effect. I really appreciated the discussion part because the authors detailed the issues of the implementation of SfM-MVS and TMEM techniques in field. The article structure is clear and readable. I think that this paper will be very useful for the coastal scientists working on the evolution of shore platforms which are common objects over many shores in the world. Moreover, protocol enhancements can be also possible such as accurate error mapping from James et al. (2017) to increase the level of detection of changes.

However, I suggest to detail more the photographs recording protocol. Indeed, do you take random oblique pictures or sub-vertical ones? This information will be very useful for non-specialist readers if they want to reproduce your protocol for their own research. Then, on field, depends on the type of rock and the degree of brightness, you can have specular reflection which lead to bad quality photograph. I suggest using diffuser to reduce and homogenise brightness over the object. This can be a suggestion into the discussion. Then, I also appreciate to have topographic profiles from DEMs and DoD graphs crossing erosion features, between stage 0 and 3 for example. This will strengthen the demonstration and the advantages of SFM-MVS photogrammetry for sub millimetre measurements. Finally, I also suggest adding the size of simulated erosion features practiced over the block experiments. I also think that there is a lack

of direct measurements of block geometry, using TMEM for example, with SfM-MVS photogrammetry results.

Globally, I have notified minor revisions, described below, for this paper.

- General forms remark: Homogenise figure calling into the text,

Line 99 - "...utilising widely available software (e.g. ArcMap, CloudCompare) for geomorphic change detection to quantify... " I suggest to replace " ArcMap " by " GIS software such as ESRI ArcGIS desktop or QGIS ".

line 102 - " ...Westoby et al. (2012), Verma and Bourke ( for more ... " close the parenthesis and indicate "in review" may be

line 153 - "...to capture different scales of erosion fromgranular scale abrasion ..." forgot a blank space

line 178 - "We used a Nikon D5500 with a variable zoom lens set up at 24 mm focal length." The focal length set at 24mm from variable zoom lens is noted into EXIF file of picture, or Photoscan estimate it?

Lines 187 – 191 – I suggest to provide the magnitude (globally) of simulated erosion processes in mm, such as deep of shallow scratch or abrasion, the size of rock block removed, etc...

Lines 204 – 206 – Do you enhance image rendering and texture using some treatments (increase contrast for example) or you just convert RAW in TIFF format?

Lines 207 – 208 - "Baseline DEMs..." Regarding non-specialist readers, I suggest to also indicate that DEMs raster grids need to be generated with common pixel coordinate origin in addition to common resolution.

Lines 290 – 306 & 315-334 & 335-352 – Results: I suggest adding, if possible, direct measurements of surface topography and simulated erosion features to compare with reconstructed scene geometry. This will be useful especially for scratch because the
magnitude of topographic changes due to this type of features can be always within the LoD.

Figures:

General remarks: I suggest using the same caption notation between figure graphic and legend

Figure 1: I suggest adding scale bar on the elements; moreover, I think that a caption showing the picture location over the surface or picture overlap map, such as some view from Photoscan, can illustrate photographs recording strategy.

Figure 2: I suggest to expand the scale of colour bar in order to observe the spatial variability of DoD between compared DEMs of control blocks.

Figure 3, 4, 5 – I suggest to eventually used the same regular interval scale in mm for DoD and after LoD in order to increase readability of legend and colours. I also suggest to plot graphs presenting DEMs between stage 0 and 3 and DoD, crossing erosion features, in order to appreciate the quality of reconstruction with SfM-MVS.

Figure 5 – The DoD after application of LoD captions in this figure display strange colour bars where the LoD limits not appear clearly.

---

## Author Comment (AC1) · 19 Sep 2018

Authors Final Response: Response to Reviewer 1 comments AC: We are grateful for the detailed and constructive critique provided by Professor Stephenson and we appreciate the time taken to review this manuscript. We have taken on board all of the comments and suggestions provided and have edited the manuscript accordingly. Responses to comments and suggested edits and provided below.

RC: This paper offers an interesting and useful comparison between SfM and the

[Figure]

TMEM methods. It helps to demonstrate how erosion measurements on shore platforms can be made across a range of scales from sub-millimetre to centimetres. While this is welcome I think the absence of a genuine field trial and data from a shore platform is a significant weakness. I suspect that such data are available and hope that some could be presented in a revised version of the manuscript to show that it actually worked in the field. AC:Response: We agree that a field trail is necessary and indeed have a project pending to undertake such a study. However we strongly disagree that this represents a significant weakeness of the paper. The paper concerns the development of the experimental protocol and testing of the approach under controlled conditions. A field study is beyond the scope of this manuscript, principally because of the time required for many of the hard-rock coastal platforms in Ireland to erode. This might help address my second main concern related to rock swelling. RC: Throughout the paper there are many places where further reference to rock swelling is needed. Certainly in the consideration of SfM versus TMEM the value of the TMEM for the study of rock swelling has not been given enough attention. AC:Response: We agree with and have included additional references to platform swelling in the manuscript as recommended. Specifics are provided in response to further comments regarding the same below. RC: In addition to swelling the TMEM has enabled much shorter timescales to be considered. At line 53 – there needs to be a correction, since short times scales are actually hours not years based on a number of studies that have investigated rock swelling over hours and days. AC:Response: We agree and have corrected the text to clarify reference to hourly measurements of platform swelling. RC: Can your SfM method detect short term variability in rock surfaces? Reference to this point is needed at line 115 needed. AC:Response: This is unlikely given the spatial scales involved. We have included additional text to highlight the resolution limitations of SfM-MVS for measuring processes such as platform swelling (Lines 126 – 129 of revised manuscript) RC: Where multiple references are provided these must be in chronological order, not alphabetical (I suspect that is an artefact of using reference management software). AC:Response: This was an error in formatting and intext citations with multiple citations have been changed throughout the text to order of publication by year (earliest to most recent). RC: It is important that chronological order is used so to recognised correct attribution and who made the contribution first. E.G. it is important that Spate et al is recognised before Moses et al 2014, since Spate clearly identified sources of error long before Moses. AC:Response: We agree and have edited the text clarify that Moses et al. (2014) refers to limitations identified by previous research and the relevant citations have been added. (lines 63- 67) in revised manuscript) RC: Scale terms need far more careful definition, micro – meso and macro scales need to be defined with a range of values. E.g. at line 165, what is micro to meso? AC:Response: We agree and have clarified this by where micro refers to sub mm and meso refers to > 1mm to cms (line 189 in revised manuscript).

RC: I wondered if it is possible to expand the applicability of the paper by reference to rock erosion more broadly than just shore platforms. This should probably include reference to: Turowski, J.M. and Cook, K.L., 2017. Field techniques for measuring bedrock erosion and denudation. Earth Surface Processes and Landforms, 42(1), pp.109-127. After all the method is not limited to shore platforms and clearly has potential in any environment where bedrock erosion is of interest. AC:Response: We agree and have inserted ne text that highlights the broader application of the method, including reference to Turowski and Cook (2017).

RC: I find the use of the term "rock shore platform" unnecessary, it is almost a tautology – just shore platform. AC:Response: We agree and used 'shore platforms' throughout the text.

RC: Line 65 The limitation to smooth surfaces is pretty well known, I don't think you can claim to have identified this limitation. AC:Response: The text has been edited to clarify this point RC: Line 142, what does "specially" mean? Needs explanations. AC:Response: We have clarified the text to indicate that 'specifically designed' refers to the design of the robust new coordinate system which is based on the coordinate system described in Verma and Bourke (2018).

RC: +/- how many millimetres? AC:Response: We have inserted the following text in Line 155 to improve clarity 'Distance/angle between targets can also be measured after application to ensure accuracy of placement and application can be repeated if necessary'.

RC: Line 148 how was the bolt levelled? Move explanation from lower down up. AC:Response: We have done this.

RC: Line 150 you need to tell the reader what is the "high degree of relocation precision". Is this same as the Kelvin clamp used by the TMEM? Or is it +/- some number and unit? AC:Response: The relocation precision is validated in the results section and we have clarified refer the reader to this in line 174 in revised document). RC: Since this is a technical methods paper I think you need to provide a technical drawing of your bolt and plate so others can manufacture both themselves. AC:Response: We agree. The bolt is a standard square head bolt available from Stig Fasteners UK (SQHM8x75) . We have requested a technical drawing of the CRS from the manufacturers and this will be available from the authors upon request as soon as it is available. We will also attach it as SOM to the field validation paper when published. Bolt manfactururs and specifications have been added to text (lines 204-205 in revised manuscript) RC: Fragments and granular need definition, what size are these? AC:Response: The size has been specified in the text. RC: Caption to Fig. 1 can you add some dimensions of the triangle angles so we get a better sense of scale? AC:Response: Scale bars have been added to the figure RC: Was the camera held by hand or placed on a tripod? Does it matter? AC:Response: We have confirmed the use of a tripod to reduce effects of hand shake on image quality (line 209 in revised manuscript) RC: Again what does "smaller-scale" mean? AC:Response: We have clarified the text on 'smaller scale processes' with examples in lines 215 in revised manuscript. RC: A key point you have not made and this becomes evident at start of the results is that SfM is an order of magnitude lower in precession than the MEM. AC:Response: We agree and now include that highlights the relatively lower resolution when compared to the

T/MEM. We emphasise that this would render the approach unsuitable for measuring change due to processes, such platform swelling, which operate at finer spatial scales. Lines 558-564. RC: Lines 368-69, actually I think the key is not factors such as cost (they are actually pretty similar if you cost the camera) but what questions are being asked and so at what scale are measurements required. AC:Response: We agree and have emphasised this in the text (Line 431-432 and 435-436) RC: Installation times are way off here, it does not take 80 minutes to install a MEM site. An experienced operator with a good drill, can install a site in 20 minutes or less. If two people are operating together it can be very fast. The other factor is the rock type, more resistant rock, then yes a bit slower. AC:Response: We have altered the time given to show the range of installation times and stated that these will depend on operator experience and rock type. (Line 440-441 and Figure 7.) RC: In section 3.3 you need to discuss the benefit the TMEM provides for investigating rock swelling, something you have not really dealt with, and not (yet) shown to be detectable by the SfM method. You need to say something about this in your discussion section as well. AC:Response: We have added text to highlight the efficacy of the T/MEM for measuring processes which operate at a higher spatial resolution that is obtainable using the SfM MVS approach (section 3.3.6 line 480-486) and note its importance for measuring of phenomena such as platform swelling in the discussion line 558-564. RC: Minor typos Line 63 Stephenson and Kirk 1966 – probably 1996 or is it 1998?? AC:Response: Corrected to Stephenson and Kirk, (1996) (line 71). RC: Cullen and Bourke 2018 not in reference list. AC:Response: Completed - Cullen and Bourke 2018 have been added to reference list. RC: Verma and Burke without year or in press. Response: Completed – year included RC: Line 62 use and between moulds and gypsum AC:Response: 'and' inserted between moulds and gypsum

Response to Reviewer 2 comments: RC: The article "A comparison of Structure from Motion Photogrammetry and the Traversing Micro Erosion Meter for measuring erosion on rock shore platforms" compare a methodological implementation of SfM-MVS photogrammetry to evaluate submillimetre erosion processes of shore platform, with

classical direct measurements from TMEM system. RC: The structure of this paper is composed of: 1) an introductory part resuming the methodologies used to measure gentle topographic changes over rock shore surface with focus on TMEM and SfM-MVS photogrammetry; 2) a detailed methodological section; 3) the results, segmented between a quality assessment of DEMs, a comparison of the level of geomorphological changes detection regarding the rugosity of surface, and a comparison between SfM-MVS photogrammetry tools and TMEM; 4) a discussion and conclusion.

RC: My opinion about this article is that it proposes a very interesting approach to quantify and map small scale erosion processes over shore platform at low cost and with accuracy. The SFM-MVS photogrammetry protocol and results are very detailed, especially regarding vertical error assessment. The results show that the method is able to reach small and medium scales detection of erosion over low rugosity surface but are clearly limited to detect loss of rock fragment over more complex topography. The SfM-MVS photogrammetry results show typical errors of geometry reconstruction associated to the technique such as shadow effect. I really appreciated the discussion part because the authors detailed the issues of the implementation of SfM-MVS and TMEM techniques in field. The article structure is clear and readable. I think that this paper will be very useful for the coastal scientists working on the evolution of shore platforms which are common objects over many shores in the world. Moreover, protocol enhancements can be also possible such as accurate error mapping from James et al. (2017) to increase the level of detection of changes.

AC:Response: We thank reviewer 2 for their detailed and constructive critique of this manuscript. We appreciate the time and effort taken to review this manuscript in such detail and are pleased with positive response. RC: However, I suggest to detail more the photographs recording protocol. Indeed, do you take random oblique pictures or sub-vertical ones? This information will be very useful for non-specialist readers if they want to reproduce your protocol for their own research.

AC:Response: We agree and now highlight that the specifics of data collection and

processing procedures are provided in the companion paper to this manuscript (Verma and Bourke (2018). RC: Then, on field, depends on the type of rock and the degree of brightness, you can have specular reflection which lead to bad quality photograph. I suggest using diffuser to reduce and homogenise brightness over the object. This can be a suggestion into the discussion. AC:Response: We agree and have included reference to lighting in the discussion. We also added the recommendations by Guidi et al. (2014), they demonstrated that use of polarising filter and digital pre-processing with HDR imaging can help to homogenise brightness over the subject subsequently improving image matching.We have included a reference to this paper in this discussion lines 535 – 537. to 475.

RC: Then, I also appreciate to have topographic profiles from DEMs and DoD graphs crossing erosion features, between stage 0 and 3 for example. This will strengthen the demonstration and the advantages of SFM-MVS photogrammetry for sub millimetre measurements. Response: We agree and include a new figure (Figure 4) with topographic profiles crossing erosion features on B1 for each stage.

RC: Finally, I also suggest adding the size of simulated erosion features practiced over the block experiments. I also think that there is a lack of direct measurements of block geometry, using TMEM for example, with SfM-MVS photogrammetry results. AC:Response: We are unclear on the suggestion provided here. As we understand reviewer 2's comment, we agree that independent measurments of erosion feature geometry would be advantages. However, we feel that the independent horizontal and vertical error checks done for each block during each stage of the experiment demonstate the accuracy of the SfM MVS reconstructions.

RC: Globally, I have notified minor revisions, described below, for this paper. RC: General forms remark: Homogenise figure calling into the text,

RC: Line 99 - "...utilising widely available software (e.g. ArcMap, CloudCompare) for geomorphic change detection to quantify... " I suggest to replace " ArcMap " by " GIS

software such as ESRI ArcGIS desktop or QGIS " (Line 113). AC:Response: We have changed this, for example, 'GIS software (e.g. ESRI ArcGIS desktop or QGIS) and other programs (e.g. CloudCompare)'

RC: line 102 - " ...Westoby et al. (2012), Verma and Bourke ( for more ... " close the parenthesis and indicate "in review" may be AC:Response: Completed changed to (year)

RC: line 153 - "...to capture different scales of erosion fromgranular scale abrasion ..." forgot a blank space AC:Response: Space inserted

RC: line 178 - "We used a Nikon D5500 with a variable zoom lens set up at 24 mm focal length." The focal length set at 24mm from variable zoom lens is noted into EXIF file of picture, or Photoscan estimate it? AC:Response: The variable zoom lens is fixed by the operator at 24mm. Note: The sensor in Nikon D5500 camera is APS-C, so the 35 mm film equivalent focal length is 36 mm.

RC: Lines 187 – 191 – I suggest to provide the magnitude (globally) of simulated erosion processes in mm, such as deep of shallow scratch or abrasion, the size of rock block removed, etc: : : AC:Response: We have included a new figure (Figure 4 with topographic profiles of erosion features on B1 in response to comments made by reviewer 2 regarding Figures 3, 4 and 5 see below). This figure illustrates the geometry of erosion features on B1 for stages 1, 2 and 3 and have included reference to this in the text (Lines 351-353). Due to the erroneous results for blocks with higher rugosity we do not use the SfM MVS reconstructions to measure geometry of erosion features on blocks 2 and 3 although we do state that the geometry of erosion features are of similar scale to B1.

RC: Lines 204 – 206 – Do you enhance image rendering and texture using some treatments (increase contrast for example) or you just convert RAW in TIFF format? AC:Response: Images were converted straight from RAW to TIFF format with no treatments were carried out on the images prior to processing. Full details of the processing

procedure used can be found in the companion paper to this manuscript (Verma and Bourke, 2018)

RC: Lines 207 – 208 - "Baseline DEMs..." Regarding non-specialist readers, I suggest to also indicate that DEMs raster grids need to be generated with common pixel coordinate origin in addition to common resolution. AC:Response: Complete common pixel coordinates are also specified in text (Line 239)

RC: Lines 290 – 306 & 315-334 & 335-352 – Results: I suggest adding, if possible, direct measurements of surface topography and simulated erosion features to compare with reconstructed scene geometry. This will be useful especially for scratch because the magnitude of topographic changes due to this type of features can be always within the LoD. AC:Response: This has been addressed with respect to comments regarding lines 187 – 191 above and Figures 3, 4 and 5 below.

RC: Figures: RC: General remarks: I suggest using the same caption notation between figure graphic and legend AC:Response: Complete - changed to same notation RC: Figure 1: I suggest adding scale bar on the elements; moreover, I think that a caption AC:Response: Complete- scale bars have been added to the figure showing the picture location over the surface or picture overlap map, such as some view from Photoscan, can illustrate photographs recording strategy. AC:Response: Figure 1 has been edited to illustrate image acquisition strategy.

RC: Figure 2: I suggest to expand the scale of colour bar in order to observe the spatial variability of DoD between compared DEMs of control blocks. AC:Response: The scale of colour bar has been expanded to show variability of DoD

RC: Figure 3, 4, 5 – I suggest to eventually used the same regular interval scale in mm for DoD and after LoD in order to increase readability of legend and colours. AC:Response: The scale bars for DoD have been changed to show regular intervals of chnage. RC: I also suggest to plot graphs presenting DEMs between stage 0 and 3 and DoD, crossing erosion features, in order to appreciate the quality of reconstruction

**ESurfD**
with SfM-MVS. AC:Response: We have included a a new figure (figure 4) showing the topographic profiles of erosion features for B1. We do not include topographic profiles of erosion feature for B2 and B3 due to the error due to rugosity discussed. RC: Figure 5 – The DoD after application of LoD captions in this figure display strange colour bars where the LoD limits not appear clearly. AC: Response: Complete - The figure has been changed.

---

## Referee Report (RR1)

The article « A comparison of Structure from Motion Photogrammetry and the Traversing Micro Erosion Meter for measuring erosion on rock shore platforms », presented by Niamh D. Cullen, Ankit K. Verma, and Mary C. Bourke, in its final version, illustrates well the potential and the limitations of SfM-MVS photogrammetry technique to detect sub-millimetres changes on objects such as shore platforms. The technique and protocol established by authors are well described. The new figures illustrate well the protocol and the results that you can expect with SfM MVS technique. The authors also provide a complete assessment of 1) scene geometry reconstruction errors and 2) of the limitations in vertical changes detection associated. Following the remarks of reviewer 1 and me, the authors straightened their demonstration in this paper. My opinion is that the paper is ready for publication. I am very confident that this type of methodological paper can find a good audience in earth sciences researcher's community and also will inspire other coastal scientists working on rocky coasts.